

# Exploring the sensitivity of the large-scale atmosphere circulation to changes in surface temperature gradients using a Statistical-Dynamical Atmosphere Model

Sonja Totz[1,2], Stefan Petri[1], Jascha Lehmann[1], Erik Peukert[1,2], Dim Coumou[1,3]

[1] Potsdam Institute for Climate Impact Research, Potsdam, Germany
[2] Department of Physics, University of Potsdam, Germany
[3] Institute for Environmental Studies (IVM), VU University Amsterdam

*Correspondence to*: S. Totz (totz@pik-potsdam.de), +49 331 288 20807

**Abstract.**

Climate and weather conditions in the mid-latitudes are strongly driven by the large-scale atmosphere circulation. Observational data indicates that important components of the large-scale circulation have changed in recent decades including the strength of the Hadley cell, jets, storm tracks and planetary waves.

Here, we use a statistical-dynamical atmosphere model (SDAM) to analyse the sensitivity of the Northern Hemisphere dynamical components to changes in temperature fields by systematically altering the zonal temperature asymmetry and meridional temperature gradient as well as the global mean temperature. Our results show that the strength of the Hadley cell, storm tracks and jet streams depends almost linearly on both the global mean temperature and the meridional temperature gradient whereas the zonal temperature asymmetry has little or no influence. The magnitude of planetary waves is clearly affected by all three temperature components. The width of the Hadley cell behaves nonlinearly with respect to all three temperature components.

Under global warming the temperature gradients are expected to change: Enhanced warming is expected in the Arctic, largely near the surface, and at the equator at high altitudes, altering the meridional temperature gradient. Further, land-ocean contrasts will change due to enhanced land warming. Also there is a pronounced seasonality to these warming patterns. Using SDAMs to disentangle and separately analyse the effect of individual temperature changes can help to understand observed and projected changes in large-scale atmosphere dynamics.

Moreover, some of these observed large-scale atmospheric changes are expected from dynamical equations and therefore an important part of model validation.

**Keywords:**



Earth System Model of Intermediate Complexity, sensitivity experiments, Statistical-dynamical atmosphere models, Hadley cell, Jet stream, Circulation changes

## 1    Introduction

Large-scale atmosphere dynamics including Hadley cells, jet streams, storm tracks and planetary waves play a key role in the general circulation of the atmosphere, determining climatic conditions worldwide. There is emerging evidence that these large-scale dynamics have significantly changed in recent observations and/or might do so in climate change projections of the 21st century.

Hadley cells are large-scale atmospheric circulations in the tropics and responsible for the transport of heat and moisture from the equator to the mid-latitudes (D'Agostino and Lionello, 2016). Recent literature suggests that the Hadley cells have widened in the past few decades due to ozone depletion in the Southern Hemisphere (Kang et al., 2011; Son et al., 2009) and due to increases in different warming agents as carbon dioxide and tropospheric ozone in the Northern Hemisphere (Allen et al., 2012). Further possible drivers of the Hadley cell dynamics are sea surface temperature variations, which may lead to tropical contraction (Allen et al., 2014), stratospheric cooling, global warming or changes in baroclinic eddy phase speeds (Chen et al., 2014). While most studies based on reanalysis products observe that together with this widening there is also a strengthening of the Hadley cells (Nguyen et al., 2013), modelling studies project a weakening over the 21$^{st}$ century under a high emission scenario (Lu et al., 2007). This discrepancy may be explained by relatively short observational records, large natural variability or model deficiencies (Allen et al., 2014).

Changes in the strength and the width of the Hadley cell circulation have strong implications for a variety of atmospheric phenomena such as jet streams, extratropical storms and planetary waves.

Jet streams are upper-level fast currents of westerly winds that circulate around the hemisphere. Under climate change, a weakening of the northern subtropical jet stream in winter is observed by most studies (Molnos et al., 2017; Rikus, 2015) though a few studies observe also a strengthening trend (Pena-Ortiz et al., 2013). The underlying drivers for those changes are still debated. One possible driver is the Arctic amplification leading to a decreasing meridional temperature gradient and a weakening of the jet stream and storm tracks.

Storm tracks play a crucial role in modulating precipitation in the Earth system (Raible et al., 2007; Hawcroft et al., 2012; Lehmann and Coumou, 2015). Yin (2005) studied storm tracks under climate change using 15 coupled climate models and found that storm tracks shift poleward and intensify under climate change in winter In addition, O'Gorman (2011) discovered that the storm track intensity is nonlinearly related to global warming. Coumou et al. (2015) show that during the



satellite period (1979-2013) storm tracks have significantly weakened in boreal summer. They argue that the weakening might be related to a decreasing meridional temperature gradient. Consistently, CMIP5 models project a further decrease in summer storm track activity under a high-emission scenario until the end of the 21st century (Lehmann et al., 2014).

In contrast to the changes in these large-scale circulation components, a recent study by Barnes (2013) indicates that there are no significant trends in the strength of long-term quasi-stationary planetary waves under climate change. These waves strongly interact with storm track activity in the mid-latitudes.

The large-scale atmospheric circulation is mainly a result of large-scale temperature differences between the equator and the poles and between the ocean and land. Different modelling studies have thus tried to examine the influences of different temperature sources on changes in mid-latitude circulation. Using an idealized general circulation model, Butler et al. (2010)
separate the temperature effects by using different heating sources. Their main findings are that a warming tropical troposphere causes poleward shifts in the extratropical tropospheric storm tracks and a weakened stratospheric Brewer–Dobson Circulation, a polar stratosphere cooling leads also to a shift of the extratropical storm tracks and a warming at the polar surface results in a equatorward shift of the storm tracks. With their approach Butler et al. can attribute which forcing has the most important influence on the shift of jet streams, storm tracks etc. In a study from 2011 Butler et al. presented an
alternative perspective on the response of the mid-latitude tropospheric circulation to zonal-mean tropical heating. The projection of the heating onto the isentropic surfaces at extratropical latitudes drive the poleward shift in wave generation at lower levels. In addition, the poleward shift in the heat fluxes within the troposphere and the diffusive nature of eddy fluxes of the polar vortex lead to a poleward shift in wave breaking near the tropospause.

In addition, Yuval and Kaspi (2016) investigated the influence of changes in the vertical structure of baroclinicity onto the
magnitude of eddy kinetic energy and eddy fluxes using an idealized global circulation model. This is especially interesting, since GCM simulations indicate that under increasing $CO_2$ concentrations the lower-tropospheric temperature gradient will decrease whereas the upper tropospheric temperature gradient will increase with counteracting effects on eddy activity (Yuval and Kaspi, 2016). The results from Yuval and Kaspi demonstrate that eddy activity is more sensitive to temperature gradient changes in the upper troposphere. In addition, recent studies using dry and most idealized general circulation
models suggest that the strength and extent of the Hadley cell depend on static stability, meridional temperature gradient and the tropopause level (Schneider and Walker, 2008; O'Gorman, 2011; Levine and Schneider, 2015).

Moreover, Shaw and Voigt (2015) examined the radiative changes of clouds and water vapour for two different aquaplanet climate models and found that they are important for the regional response of precipitation and atmosphere circulation. They concluded that uncertainty in circulation is linked to uncertainty in the behaviour of clouds and water vapour.

However, none of the above mentioned studies analyzed the impact of zonal temperature asymmetry and meridional temperature gradient as well as global mean temperature by changing the temperature components directly without adding a heat source or increasing $CO_2$. For that reason, changes in the Hadley cell, jet streams, planetary waves and storm tracks



might not be directly assigned to individual temperature components. If possible, we compare our results with the literature as part of of model validation.

For the analysis presented here we use the statistical-dynamical atmosphere model Aeolus 1.0 (Totz et al., 2018), which is explained in section 2 in more detail. In section 3, we describe the data used for the experiments, the separation of the

temperature components and the analysed dynamical variables. In section 4 we present the results and in section 5 compare them with the literature. In section 6 we provide a short discussion on the robustness and interpretation of our results. We conclude with a summary in section 6.

## 2    Model description

The experiments are run with the statistical-dynamical atmosphere model Aeolus 1.0 (Coumou et al., 2011; Totz et al., 2018). It is a 2.5 – dimensional model with the vertical dimension coarsely resolved and therefore belongs to the model class of intermediate complexity atmosphere models (Claussen et al., 2002; Petoukhov et al., 2000). Aeolus is based on time-averaged (over short time scales) equations in which transient eddies are parameterized in terms of the large scale field (Peixoto and Oort, 1992; Saltzman, 1978). This means that instead of solving every eddy directly, only the ensemble mean

eddy characteristics (in terms of heat, water vapour and momentum transport) are solved. The essential difference compared to more widely used general circulation models (GCMs) is thus the point of truncation in the frequency spectrum of atmospheric motion (Saltzman, 1978). This different approach allows much coarser spatial and temporal discretization, making SDAMs computationally efficient, because the synoptic waves are parameterized in terms of the large scale wind field which is the basic idea of a statistical-dynamical method. The model includes the two-way dynamical interactions

between parameterized storms and resolved westerlies. A full description can be found in Coumou et al. (2011).

The model has 5 vertical levels in the troposphere with the model top at 10000m altitude. Aeolus 1.0 has a "dummy" stratosphere (i.e. its physics and dynamics are not resolved) to have a boundary condition at the top of the troposphere. In this experiment we excluded topographic influences and it is an atmosphere-only setup using prescribed sea level temperatures.

Aeolus 1.0 contains a full hydrological cycle consisting of three-layer stratiform cloud plus convective cloud scheme as presented and validated in Eliseev et al. (2013). The convective plus 3-layer stratiform cloud scheme includes low-level, mid-level and upper-level stratiform clouds. The equation for humidity is a prognostic equation and described in Petoukhov et al. (2000).

Parameter values for the dynamical core were taken from the calibration process, as described in Totz et al. (2018), which

optimizes the model's representation of the tropospheric large-scale circulation.

For more information such as comparisons of Aeolus 1.0 with GCMs or the equations for planetary waves, zonal mean



meridional and zonal mean velocities as well as the azonal wind velocities, we refer the reader to Totz et al. (2018).

## 3  Data and Methods

### 3.1  Aeolus forcing parameters

The simulations are forced by climatological (1979-2014) winter mean (December-January-February (DJF)) data of sea surface temperature and specific humidity at the surface, using ERA-Interim data from the European Centre for Medium-Range Weather Forecasts (ECMWF) (Dee et al., 2011). First, the data is regridded to $3.75° \times 3.75°$ (longitude $\times$ latitude). In this experiment, temperature and humidity are prescribed to decouple the dynamics from diabatic heating and associated temperature changes. This way, the dynamical core equilibrates to the prescribed temperature and humidity patterns without

any additional complicating factors. We then test how different changes in the temperature profiles affect different aspects of the circulation. We do thousands of individual simulations to disentangle and separately analyse the effect of global mean temperature, equator-to-pole temperature gradient and east-west temperature differences.

### 3.2  Specifications of the surface temperature

In this and the following sections, the angle brackets denote time-averaged quantities, the overbar denotes zonal mean quantities, the prime indicates synoptic scale components (2 – 6 days period), and the star indicates azonal components, i.e. deviations from the zonal mean.

For the sensitivity analysis we vary three different temperature components: (1) the meridional temperature gradient $\frac{dT}{d\phi}$, (2) the zonal temperature asymmetry $T^*$ (i.e. deviations from the zonal mean) and (3) the global mean temperature $T_{global}$.

We change the temperature for each grid cell with respect to parameters for the three components in three steps. First, the parameter $w_{T_\phi}$ is used to vary the meridional temperature gradient by cooling/warming the poles

$$T_1(\phi,\lambda) = T_{EQ}(\lambda) + \left( T_{DJF}(\phi,\lambda) - T_{EQ}(\lambda) \right) * w_{T_\phi}, \tag{1}$$

whereby $\phi$ and $\lambda$ are respectively latitude and longitude, $T_{DJF}(\phi,\lambda)$ is the original temperature, $T_{EQ}(\lambda)$ the temperature at the equator and $T_1(\phi,\lambda)$ is the altered temperature. In temperature $T_1$ only the meridional temperature gradient is altered/ updated and the temperature change of the azonal component as well as the global mean temperature are done in the next

steps.

In the second step the global mean temperature is adapted to the originally global mean temperature $T_{DJF}$ and global mean temperature is varied by the parameter $T_{global}$



$$T_2(\phi, \lambda) = T_1(\phi, \lambda) - \left( \mathrm{Mean}(T_1) - \mathrm{Mean}(T_{DJF}) \right) + T_{global}, \qquad (2)$$

Whereby $\mathrm{Mean}(T_1)$ is the global mean temperature of $T_1$ and $\mathrm{Mean}(T_{DJF})$ is the global mean temperature of $T_{DJF}$.

In the third step, the parameter $w_{azonal}$ is used to alter the azonal temperature, which is added to the zonal mean temperature $\overline{T_1}$.

$$T_{\mathrm{Final}}(\phi, \lambda) = T_2{}^*(\phi, \lambda) * w_{azonal} + \overline{T_2}(\phi) \qquad (3)$$

This way $w_{T_\phi} = 1$, $w_{azonal} = 1$ and $T_{global} = 0$ indicate present day conditions. The temperature perturbation $T_{\mathrm{Final}}$ is the

final temperature based on all three temperature components and is used as model input to which the dynamical core equilibrates. The temperature perturbations are applied at sea level and propagate to the upper levels based on the lapse rate equation. A schematic plot of the different temperature perturbations is shown in Fig. 1.

The parameters $w_{T_\phi}$ and $w_{azonal}$ are varied between 0.75 and 1.1 (with steps of 0.025) to examine the behavior of the dynamical core under conditions between $-25\%$ to $+10\%$ of their present-day wintertime climatological values. These

limits roughly correspond to expected temperature gradients during the last glacial maximum scenario and in a $2 \times CO_2$ scenario (Coumou et al., 2011). Large zonal temperature differences, i.e. large values of $w_{azonal}$, imply strong temperature deviations between land masses and oceans. Small $\frac{dT}{d\phi}$ values represent amplified warming of the poles, compared to the equator, and thus a reduced meridional temperature gradient.

The parameter $T_{global}$ is altered between $-4K$ and $+4K$ (with steps of $4K$) relative to the climatological present-day (PD)

temperature (1979 – 2014). This range covers climate variability over the past million years and possible near future changes.

For each temperature component we determine its influence on the strength and width of the Hadley cell, as well as the strength of zonal-mean jets, storm tracks and planetary waves in the Northern Hemisphere.

To force the Aeolus dynamical core, we use perturbed surface temperature profiles derived from the ERA-Interim winter climatology as explained above. We perform 2025 simulations with a regular 3-dimensional parameter space using the

multi-run simulation environment *SimEnv*, which provides a tool to inspect the model's behaviour in the parameter space by discrete numerical sampling (Flechsig et al., 2013).

### 3.3    Dynamical variables

To obtain the strength of the jet stream for this analysis, we use seasonally (DJF) averaged zonal mean zonal wind $\overline{\langle u \rangle}$. For

simplicity, we define the jet stream strength as the maximum of $\overline{\langle u \rangle}$ between 10°N and 80°N at 9000 m height (corresponding to a pressure level of ca. 300 mbar).




We define the strength of the Hadley cell as the maximum absolute zonal mean integrated mass flux between 0° and 90° latitude and the width as the distance between the mass flux zero-crossings near 0° and 30° latitude. We calculate the width and the strength of the Hadley cell by computing the integrated southward mass flux in the lower troposphere between 1000 mb and 500 mb from the zonal mean meridional wind velocity (Totz et al. , 2018).

As a measure of storm track activity we calculate the eddy kinetic energy ($E'_K = 0.5(u'^2 \pm v'^2)$) , whereby $u'$ and $v'$ are the zonal and meridional synoptic wind velocity. We average the eddy kinetic energy over all five pressure levels and calculate the maximum between 10° N and 80° N to analyse the strength and shift of the storm track activity.

We calculate the strength of the planetary waves by averaging all positive values between 20°N−80°N of the azonal wind components $\langle u^* \rangle$ and $\langle v^* \rangle$.

**4   Results**

We compare and analyze the zonal mean dynamical variables of eddy kinetic energy $\overline{\langle E'_K \rangle}$ (which captures storm track activity), zonal mean zonal wind velocity $\overline{\langle u \rangle}$ and the vertical integral of the lower tropospheric integrated mass flux $\overline{\langle m \rangle}$ as well as azonal wind velocities $\langle u^* \rangle$ and $\langle v^* \rangle$.

**4.1   Tropical circulation**

**4.1.1   Strength and width of the Hadley cell**

The integrated mass flux in the lower troposphere of the present-day modelled climatological NH winter values (black line in Fig. 2) captures well the shape of the red curve from ERA-Interim data. In particular, the maximum strength, defined as the

minimum between the zero-crossings, is close to the ERA-Interim data. There exist bigger differences in the SH. This model bias might be related to the missing Antarctic ice sheet, upper-tropospheric ozone, the constant lapse rate assumption, or fundamental limitations of the equations.

Zero-crossing refers to the point in the graph where the function $f(x)$ crosses the $y = 0$ line. The modelled Hadley cell's width, defined as the distance between the mass flux zero-crossings near 0° and 30° latitude, is smaller than in ERA-Interim.

For further analysis we plot the width (Fig. 3) and strength (Fig. 4) of the Hadley cell as a function of $w_{T_\phi}$, $w_{azonal}$ and $\Delta T_{G,PD}$, whereby $\Delta T_{G,PD}$ is the difference between the present day global mean temperature and the altered global mean temperature. In general, both a stronger meridional temperature gradient and a stronger zonal temperature contrast lead to a nonlinear broadening of the Hadley cell width: For meridional temperature gradients smaller than today, the width is smaller



for larger global mean temperature. The Hadley cell expands as well for stronger meridional temperature gradient and even for a greater zonal temperature asymmetry, but with a smaller rate.

For meridional temperature gradients larger than today, the influence of global mean temperature and meridional temperature gradient are less significant, and the values of the zonal temperature

asymmetry play a bigger role. In this case larger values of the zonal temperature asymmetry lead to a larger width of the Hadley cell. However, depending on the global mean temperature an increase of the zonal temperature asymmetry can also lead to a decrease of the Hadley cell width, e.g. $\Delta T_{G,PD} \geq 3$, $0.95 \geq w_{T_\phi} \geq 0.9$ and $w_{azonal} \geq 0.85$.

The Hadley cell width shows larger changes in response to changes in the meridional temperature gradient than for changes in the zonal temperature asymmetry, indicating that the former has a stronger relative influence.

The Hadley cell strengthens with increasing meridional gradient and depends stronger on global mean temperature than on $w_{azonal}$ (Fig. 4).

## 4.2   Extratropical circulation

### 4.2.1   Strength of the jet stream

The jet stream locations and strengths for both hemispheres are visible as two distinct maxima in the zonal-mean zonal wind velocity in ERA-Interim (Fig. 5a). Aeolus reasonably reproduces the main jet stream features in terms of spatial position and magnitude (Fig. 5c). The modelled magnitude of the jet in the Northern Hemisphere is in better agreement with reanalysis data than in the Southern Hemisphere. This is likely related to the fact that Aeolus is not coupled to an ice model and thus effects from the Antarctic ice sheet are not considered. The model reasonably reproduces near-surface tropical easterlies

("trade winds") at low latitude.

Fig. 5(b) and Fig. 5(d) show the impact of changes in the meridional temperature gradient $\frac{dT}{d\phi}$ on jet stream dynamics. With a higher meridional temperature gradient, the strength of the jet stream increases and with a lower temperature gradient the strength decreases.

This is also observed in Figure 6 where the jet stream strength is shown as function of $w_{T_\phi}$, $w_{azonal}$ and $\Delta T_{G,PD}$.

The strength of the jet stream is sensitive to the meridional temperature gradient and to the global mean temperature. The zonal temperature contrasts have little influence on the jet stream strength.

### 4.2.2   Strength of the storm track activity

The NH winter climatological (1979 - 2014) storm track's activity in Aeolus (Fig. 7(c)) is similar to ERA-Interim data (Fig. 7(a)).

Figures 7(b)-(d) show that storm track activity increases with increasing temperature gradient.

The strength of the storm track activity depends on all three components $w_{T_\phi}$, $w_{azonal}$ and $\Delta T_{G,PD}$ (Fig. 8) in a way that the

influence of $w_{T_\phi}$ dominates the influence on storm track activity. The increased global mean temperature leads to a general strengthening of the storm track activity (Fig. 8).

### 4.2.3    Strength of the planetary waves

The strength of the planetary waves is roughly as sensitive to $w_{T_\phi}$ as to $w_{azonal}$, both in terms of $\langle u^* \rangle$ (Fig. 9(a)) and in

terms of $\langle v^* \rangle$ (Fig. 9(b)). Both meridional and zonal wind directions exhibit the same relationship such that larger meridional and zonal temperature asymmetries lead to stronger winds. In addition, if the meridional temperature gradient is smaller than the zonal temperature asymmetry, the strength of the planetary waves increases faster with higher meridional temperature gradient then if both have a similar magnitude. Moreover, if the zonal temperature asymmetry is smaller than the zonal temperature asymmetry, the strength of the planetary waves increases faster with higher zonal temperature asymmetry then if

both have a similar magnitude. This behaviour leads to a curved structure of the azonal winds.

The global mean temperature has a positive but only weak influence on the strength of planetary waves.

## 5    Discussion

For all investigated atmosphere variables we observe a strengthening for higher global mean temperature and higher absolute meridional temperature gradients and only a weak (strong) dependence on the zonal temperature asymmetry for storm tracks (planetary waves and Hadley cell width), which we discuss in comparison with results from literature in the following sections. However, most previous studies have analysed only the combined effect of changes in several temperature components making a direct comparison difficult.

### 5.1    Tropical circulation

### 5.1.1    Strength and width of the Hadley cell



The strength of the Hadley cell depends strongly on the meridional temperature gradient with a stronger Hadley circulation for larger meridional gradient (Fig.4). Its strength is much less sensitive to global mean temperature explainable by the enhanced latent heat release under warmer conditions. Finally, it is almost insensitive to zonal temperature asymmetries.

In addition, our analysis suggests that the Hadley cell width depends nonlinearly on all three temperature components (Fig. 3). The dependence of the Hadley cell width on the meridional temperature gradient is consistent with findings from Frierson et al. (2007) using an idealized moist GCM and a full GCM.

However, our findings regarding the width of the Hadley cell are more complex than from Frierson et al. (2007), who only observe an increase of the width with increasing global mean temperature.

Also Adam et al. (2014) examined the Hadley cell using 6 reanalysis datasets, 22 Atmospheric Modeling Intercomparison Project (AMIP) simulations, and 11 historical Ocean-Atmosphere coupled simulations from phase 5 of the Climate Modeling Intercomparison Project (CMIP5) and observed the same behaviour for the meridional temperature gradient but the opposite behaviour for a stronger global mean temperature compared to our results. To distinguish between meridional temperature gradient and global mean temperature, they decomposed sea surface temperature (SST) into factors that are primarily associated with global warming (mean SST changes) and ENSO (SST gradient changes). They concluded that a weakening of the temperature gradient and an increase of the global mean temperature leads to a widening of the Hadley cell.

In agreement to our results, Mitas and Clement (2005) detected a strengthening of the Hadley cell in their analyses using several reanalysis data sets, a rawsonde data set and a model data set. However, they found great differences between different data sets.

Lu et al. (2007) found a robust weakening and a poleward expansion of the Hadley circulation in response to increased GHG forcing in simulations of the 21st century climate taken from the A2 scenario of the IPCC AR4 project. Lu et al. (2008) analysed the change in the zonal mean atmospheric circulation under global warming in comparison with the response to El Niño forcing, by examining the CMIP5 model simulations. They used again the A2 scenario to simulate global warming. Under global warming due to higher $CO_2$ concentrations the Hadley cell weakens and expands northwards together with a poleward shift of the jet stream. Based on our results, we can assume that "El Niño–like" enhanced warming leads to a stronger zonal temperature asymmetry (and a higher global mean temperature) resulting in a stronger Hadley cell, whereas the $CO_2$ concentration leads to a weaker meridional temperature gradient (and a higher global mean temperature) and as a consequence the Hadley cell weakens. This can also explain the widening of the Hadley cell, which we observe in our experiments as well: A decreased meridional temperature gradient for warmer global mean temperature than today can lead to a smaller width of the Hadley cell and vice versa.

Seo et al. (2014) investigated possible drivers of the Hadley cell such as the meridional temperature gradient, gross static stability and tropopause height using CMIP5 climate models. Consistent with our results, they found a robust dependence between meridional temperature gradient and the strength of the Hadley cell in winter: A decreased meridional temperature gradient leads to a weakening of the Hadley cell.





In addition, D'Agostino et al. (2017) analysed and compared the Hadley cell during the last glacial maximum to global warming scenarios (RCP4.5 and RCP8.5) with a focus on the dependence on subtropical stability, near-surface meridional potential temperature gradient, and the tropical tropopause level. They concluded that the meridional temperature gradient is a major driver for Hadley cell changes.

However, in both studies the atmospheric composition in terms of anthropogenic aerosols is changed and hence not only the meridional temperature gradient changes but also the global mean temperature and the zonal temperature asymmetry. This makes it difficult to attribute changes in the Hadley cell to one temperature component only.

## 5.2    Extratropical circulation

### 5.2.1    Strength of the jet stream

We show that the strength of the jet stream decreases with decreasing absolute meridional temperature gradient (Fig. 5 and Fig. 6).

This is in agreement with findings from Polvani and Kushner (2002) and Haigh et al (2005). Polvani and Kushner used a simple general circulation model and showed that for sufficiently strong cooling of the polar winter stratosphere, jet streams

weaken and shift poleward. Haigh et al. (2005) analyse the weakening and shift of the subtropical jet using a multiple regression analysis of the NCEP-NCAR reanalysis zonal mean zonal wind velocity. Furthermore, they show with a simple general circulation model that the generic heating of the lower stratosphere tends to weaken the subtropical jets.

In most observational studies, a weakening of the jet is observed over the last decades like Archer & Caldeira (2008) using NCEP and ERA-40 reanalysis sets, Rikus (2015) using ERA-40 data and Molnos et al. (2017) using ERA-Interim data.

However, Pena-Ortiz et al. (2013) found that trends in both strength and position of the jet strongly vary between different reanalysis products.

### 5.2.2    Strength of the storm track activity

In this study, we observe a strengthening of storm track activity under increased global-mean temperature.

Our results are supported by findings from McCabe et al. (2001), who observe a strengthening of the storm track activity with higher global temperature induced by GHG forcing. This is also in agreement with Yin (2005), who investigated 15 coupled climate models and showed that storm tracks intensify under global warming. In addition, Chang et al. (2012) found that storm tracks in the upper troposphere increase in winter using 23 CMIP5 models (below 300mb they found a slight decrease) and the strength of storm track activity depends strongly on the meridional temperature gradient, which is

consistent to our findings.



This latter result is intuitive as the prime role of storm tracks within the general circulation is to transport heat poleward, with a stronger temperature contrast leading to enhanced heat transport. It also directly follows from the equation of eddy kinetic energy, which in the first place depends on the meridional temperature gradient (Coumou et al., 2011).

Harvey et al. (2013) observe similar results using CMIP5 data: Larger temperature differences in the equator-to-pole
temperature at upper- and lower-tropospheric levels lead to stronger storm activity.

In reanalysis data also a strengthening of the storm tracks can be observed (Schneidereit et al., 2007; Wang et al., 2006), which is supposedly because of the rising global mean temperature. The zonal temperature asymmetry could not be responsible for a strengthening, since the zonal temperature asymmetry should be reduced in winter due to global warming. According to our results, this would lead to a weakening of the storm track activity.

O'Gorman and Schneider (2008) examined the response of storm tracks to different climate conditions simulating an aquaplanet and by changing the longwave optical thickness in the radiation scheme of the GCM (representing variations in greenhouse gas concentrations). They found that eddy kinetic energy has a maximum for a climate with the global-mean temperature similar to that of present-day-climate. Lower or higher global-mean temperatures lead to significantly smaller values. In addition, they observed that the eddy kinetic energy increases monotonically with the meridional insolation
gradient (representing changes in, for example, high-latitude surface albedo).

Similarly, Pfahl et al. (2015) investigated the behaviour of extratropical cyclones under strongly varying climate conditions using idealized climate model simulations in an aquaplanet setup. They changed the meridional insolation gradient together with the longwave optical thickness with shortwave parameters held constant. They found that the maximum of eddy kinetic energy is reached at a global mean temperature slightly warmer than present-day climate.

These results are different to our results, where no such peak in Eddy kinetic energy can be observed. The different results may be explained by the different techniques applied to simulate higher global mean temperature. In our study, we directly change the temperature, whereas Pfahl et al. change the longwave optical thickness with shortwave parameters held constant, which represents variations in longwave absorbers like carbon dioxide and water vapour. These changes could also change the meridional and zonal temperature asymmetry leading to different results.

Nevertheless, we also observe a strong positive dependence between temperature gradient and Eddy kinetic energy.

### 5.2.3   Strength of planetary waves

In our analysis the strengthening of the planetary waves depends on all temperature components. Larger meridional and zonal temperature asymmetries as well as global mean temperatures lead to stronger winds.

Since azonal wind components emerge due to zonal temperature asymmetries, it is expected that higher zonal temperature differences lead to stronger azonal wind components. Stronger temperature gradients cause stronger meridional wind velocities, which are deflected by the Coriolis force and therefore also the zonal wind velocities are stronger. Those wind velocities are slowed down or accelerated due to topography, land-ocean-contrast and hence also the azonal component of



the wind velocity will be stronger. In addition, a higher global mean temperature leads to more available energy in the atmosphere and therefore to a larger azonal wind velocity.

Under climate change the global mean temperature increases whereas the meridional temperature is expected to decrease. Our results suggest that this will have contrary effects on the strength of planetary waves.

Thus, this could explain the results, which Barnes (2013) found by analysing the planetary waves with wave number 1 - 6 as well as wave numbers 1 - 3. She concluded that there is no significant trend in terms of the strength of the planetary waves and thus Arctic amplification does not play a dominant role for changing the undulations of the jet stream.

## 6    Interpretation and robustness of the model results

The large-scale dynamical atmospheric changes due to variations of the temperature components, as presented here, coincide with those expected from dynamical principles, with the possible exception of the Hadley cell width changes. This therefore first of all serves as an important validation of our SDAM atmosphere model. To our knowledge, no other SDAM model exists that captures these dynamical interactions between the large-scale circulation components of tropical circulation, jets, storm tracks and planetary waves.

The fine sampling resolution of the parameter space reveals that most variables have a linear relationship to the investigated temperature components. In retrospect, a coarser resolution sampling of the parameter space using a GCM might have sufficed to detect the relations between temperature components and dynamical variables.

However, only through our many runs using the fast SDAM this linearity could be shown.

## 7    Conclusion

In this paper, we present a study on multiple fundamental components of the large-scale atmosphere dynamics to different surface temperature forcing with the statistical-dynamical Atmosphere model Aeolus 1.0. Due to the statistical-dynamical approach, Aeolus 1.0 is much faster than GCMs, which allows us to do 1000s of individual simulations and thus test the 25   sensitivity of the dynamical fields to different surface temperature changes. This way one can disentangle and separately analyse the effect of global mean temperature, equator-to-pole temperature gradient and east-west temperature differences.

The model's climatology generally reproduces the dynamical fields of ERA-Interim, especially in the Northern Hemisphere, which is the focus of our analysis. If possible, we compare our findings with results of the literature and conclude that most modelled changes are in line with theory and simulations.

These results serve as an important validation of the dynamical core of the Aeolus. In future work we would like to use the gained knowledge to simulate only specific temperature component configurations to further explore the dependence of the different atmospheric large-scale circulations on near-surface temperature profiles.

**Code and data availability**

All original data was downloaded from public archives. Code and data are stored in PIK's long term archive, and are made available to interested parties on request.

**Author Attribution**

S. Totz, E. Peukert and D. Coumou developed the study Conception. S. Totz, S. Petri, E. Peukert and D. Coumou developed the analysis method. D. Coumou, S. Petri and S.Totz developed the model code and performed the simulations. S. Totz, J. Lehmann
and D. Coumou analysed and interpreted the data. S. Totz wrote the manuscript with contributions from all co-authors.

**Team list**

S. Totz, S. Petri, J. Lehmann, E. Peukert, D. Coumou

**Competing interests**

The authors declare that they have no conflict of interest.

**Acknowledgements**

We thank ECMWF for making the ERA-Interim available. The work was supported by the German Federal Ministry of Education and Research, grant no. 01LN1304A, (S.T., D.C.). The authors gratefully acknowledge the European Regional Development Fund (ERDF), the German Federal Ministry of Education and Research and the Land Brandenburg for supporting this project by providing resources on the high performance computer system at the Potsdam Institute for Climate
Impact Research.






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



**Figures**

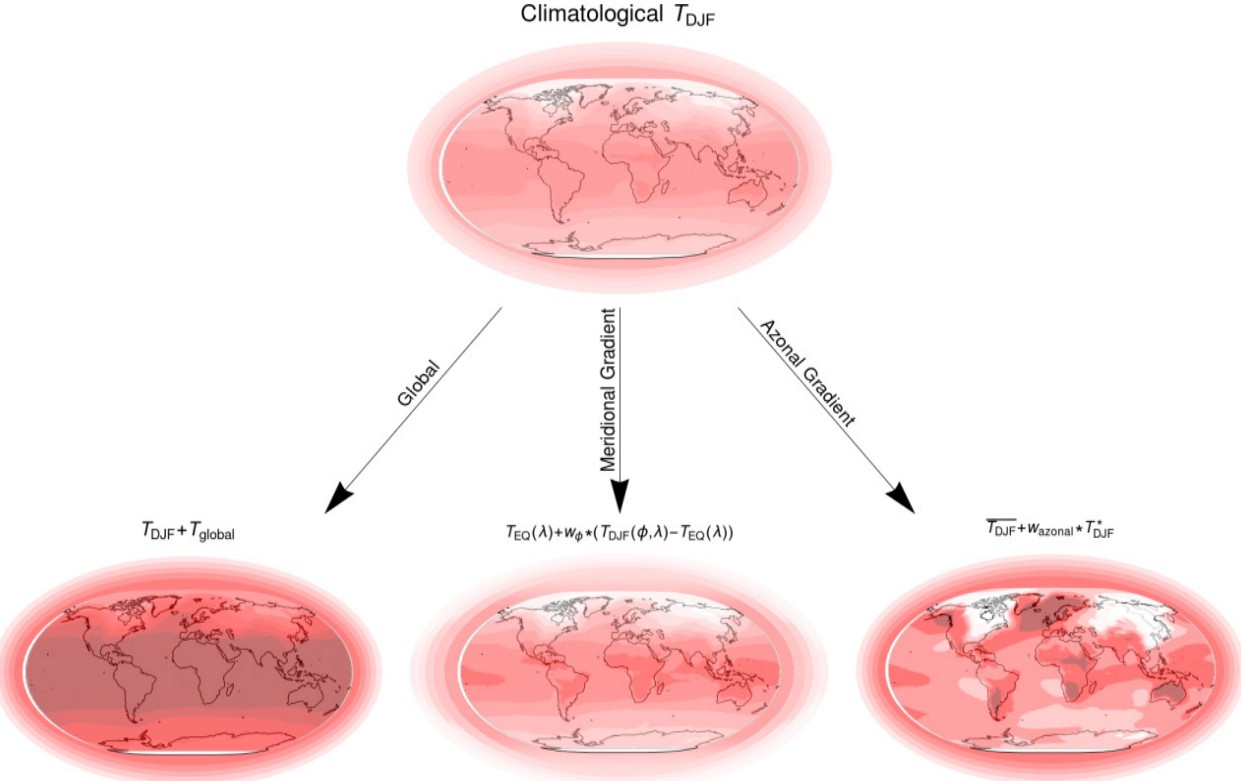

**Fig. 1** Schematic illustration of the temperature perturbations.

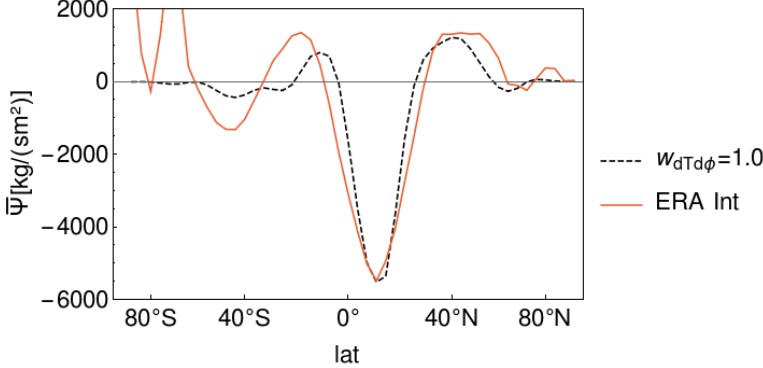

**Fig. 2** Integrated northward mass flux in lower troposphere. Black: Model output for $w_{T_\phi} = 1$ and $w_{azonal} = 1$. Red: ERA Interim climatological winter data.




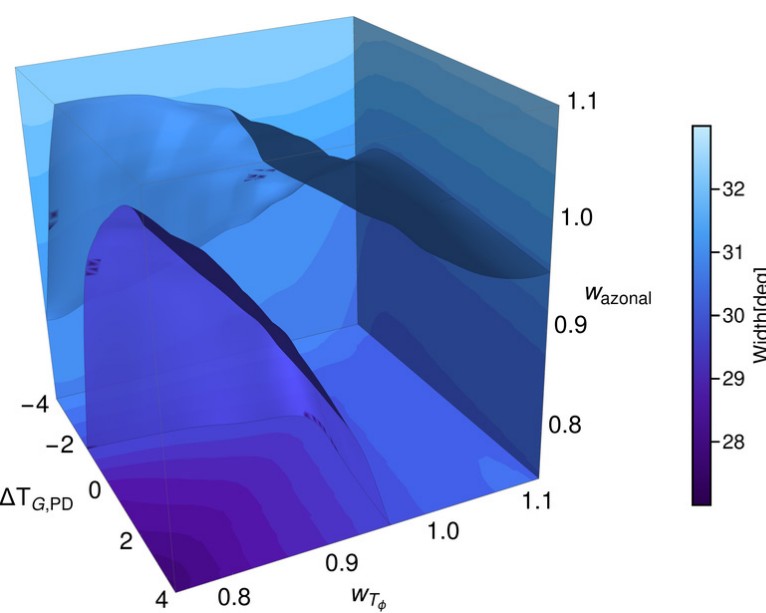

**Fig. 3** Width of the Hadley cell in dependence of $w_{T_\phi}$ and $w_{azonal}$ and $\Delta T_{G,PD}$, whereby $\Delta T_{G,PD}$ is the difference between the present day temperature and the altered global mean temperature.

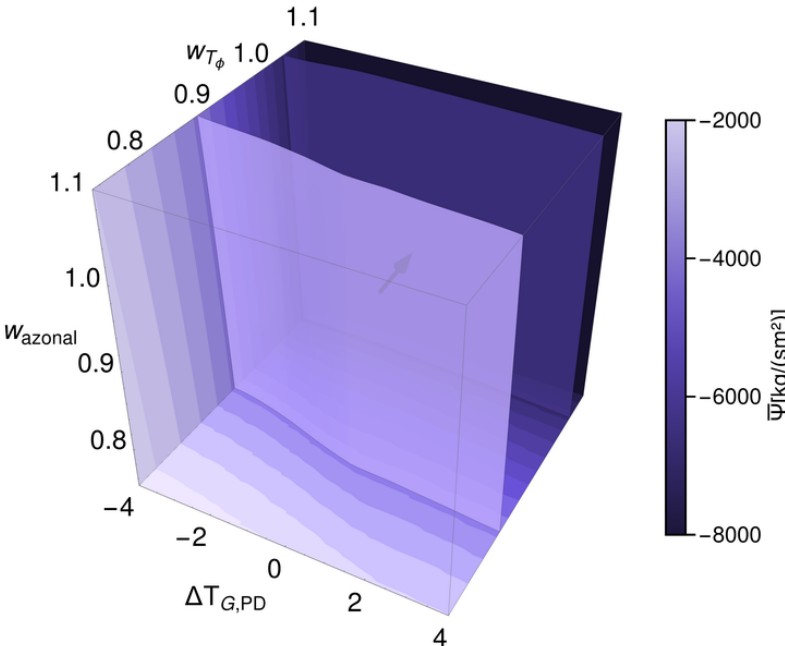

5    **Fig. 4** Strength of the Hadley cell in dependence of $w_{T_\phi}$ and $w_{azonal}$ and $\Delta T_{G,PD}$. The arrow points in the direction of the strongest gradient.



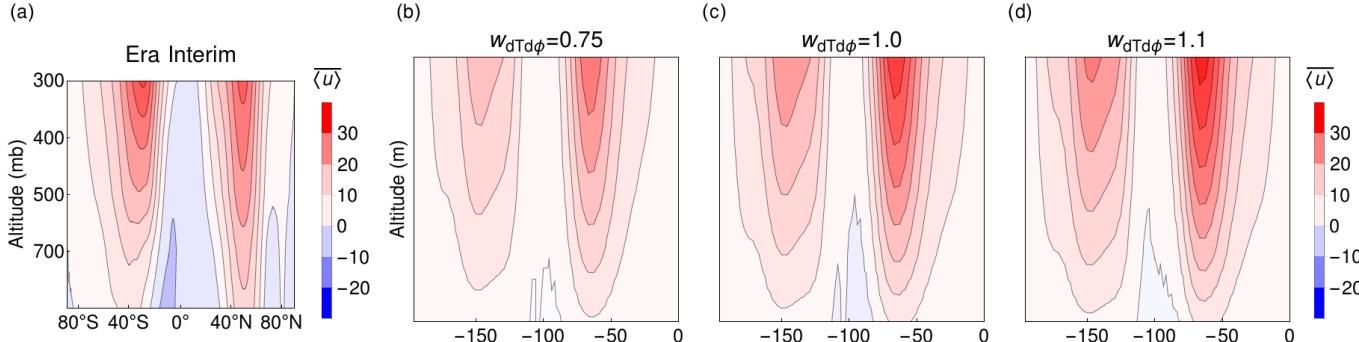

**Fig. 5** Zonal mean zonal wind velocity $\overline{\langle u \rangle}$ representing the jet streams. The subpanel (a) shows ERA-Interim data, the others results from Aeolus with different $\boldsymbol{w_{T_\phi}}$. In (b) $\boldsymbol{w_{T_\phi} = 0.75}$, in (c) $\boldsymbol{w_{T_\phi} = 1.0}$ and in (d) $\boldsymbol{w_{T_\phi} = 1.1}$. All other parameters are set to standard values.

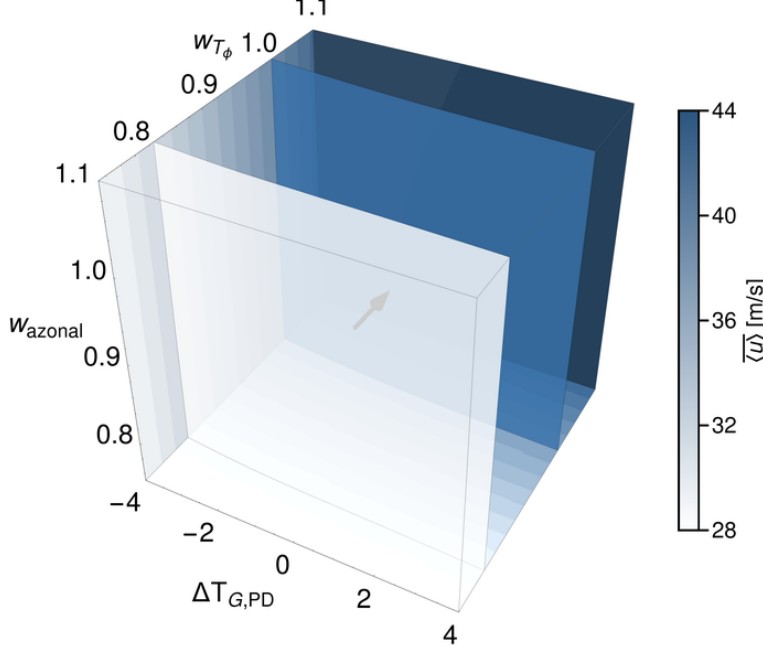

**Fig. 6** Jet stream strength defined by the maximum of the zonal mean zonal wind velocity ( $\overline{\langle u \rangle}$ ) between 10°N and 80°N at a height of 9000 m (corresponds to 300mb) in dependence of $\boldsymbol{w_{T_\phi}}$ and $\boldsymbol{w_{azonal}}$ and $\boldsymbol{\Delta T_{G,PD}}$.




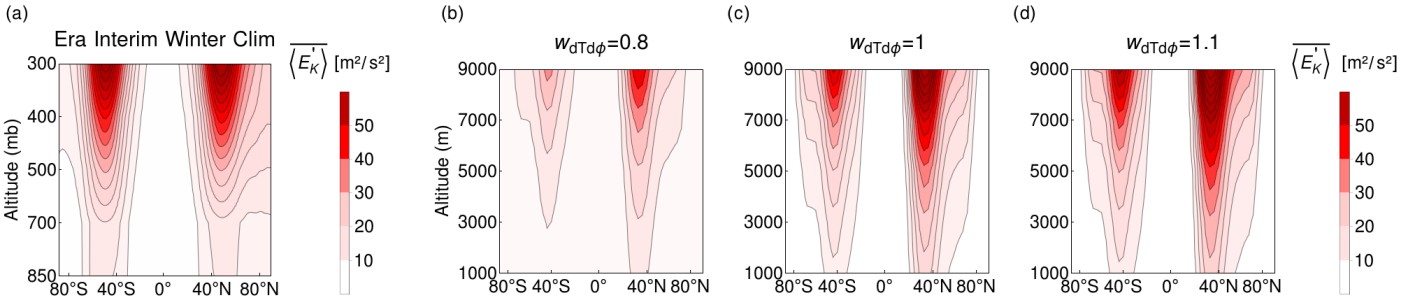

**Fig. 7** Eddy kinetic energy representing storm track activity. Panel (a) shows ERA Interim data, and (b)-(d) from Aeolus with different poleward temperature gradients $w_{T_\phi}$. In (b) $w_{T_\phi} = 0.75$, in (c) $w_{T_\phi} = 1.0$ and in (d) $w_{T_\phi} = 1.1$. All other parameters are set to climatology. With larger gradients the storm track acitivity gets stronger.

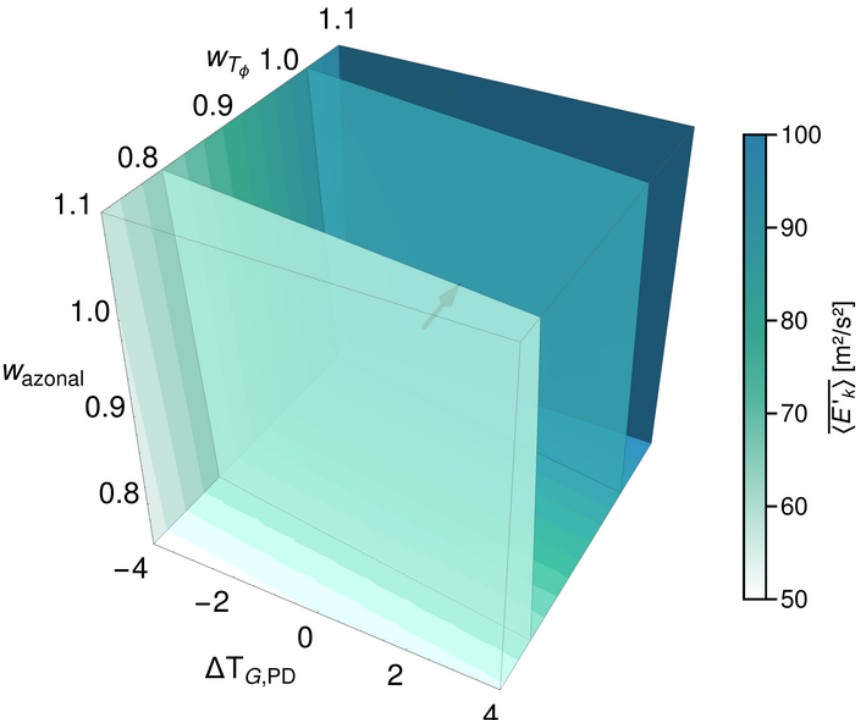

**Fig. 8** Strength of the vertical integrated storm track activity in dependence of $w_{T_\phi}$ and $w_{azonal}$ and $\Delta T_{G,PD}$. The arrow points in the direction of the strongest gradient.





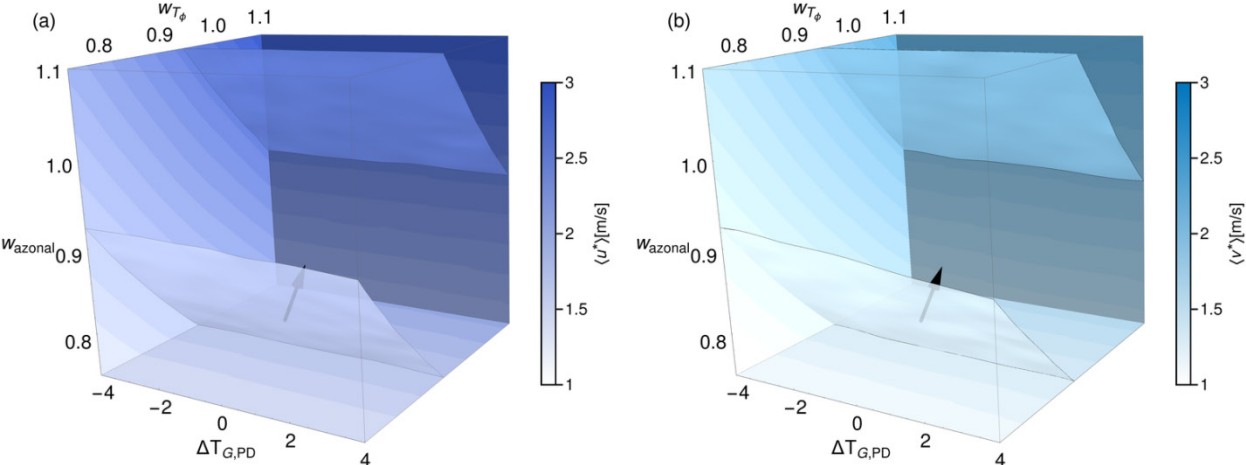

**Fig. 9** Strength of planetary waves $\langle u^* \rangle$ and $\langle v^* \rangle$ in dependence of $w_{T_\phi}$ and $w_{azonal}$ and $\Delta T_{G,PD}$. The arrow points in the direction of the strongest gradient.

