# Peer review of "Exploring the sensitivity of Northern Hemisphere atmospheric circulation to different surface temperature forcing using a Statistical Dynamical Atmospheric Model"

_Nonlinear Processes in Geophysics, 2018_

## Referee Comment (RC1) · Anonymous Referee #1 · 7 Jul 2018

Review of "Exploring the sensitivity of the large-scale atmosphere circulation to changes in surface temperature gradients using a Statistical-Dynamical Atmosphere Model " Sonja Totz, Stefan Petri , Jascha Lehmann, Erik Peukert, Dim Coumou

Referee #1

The paper investigates the role of global mean, meridional and azonal temperature changes on large scale Northern Hemisphere atmospheric circulation (jet stream, storm track, planetary wave and Hadley Circulation) within properly designed simulations performed with a statistical-dynamical atmosphere model (SDAM) Aeolus 1.0. The authors found that the strength of the Hadley cell, storm tracks and jet streams depends almost linearly on both the global mean temperature and the meridional temperature gradient whereas the zonal temperature gradient has little or no influence. The width of the Hadley cell behaves nonlinearly with respect to all three temperature components. In spite of the well structured manuscript (in terms of sections) the story flow is still poor and sometimes the main aim is unclear through the reading. Hence, I recommend a major revision. See major and specific comments below.

Major Comments:

After the first rejection in ESDD, the new manuscript addressed pretty well the points made by reviewers in the previous submission. Authors now described very precisely the model and the model set-up used for the study, highlighting the strong points of the Statistical-dynamical Atmospheric Model SDAM-Aeolus 1.0 with respect to conventional CGMs used to investigate Hadley Cell (HC) dynamics and its changes. This represents the novelty of this study.

However, it is very hard to find the main goal of the paper: in the beginning, disentangling the effect of changes in the meridional temperature gradient versus azonal temperature changes versus mean temperature changes on the boreal winter atmospheric circulation seems to be the main goal of the study. In the second part of the paper however, it seems that authors want to validate only the SDAM in order to assess its performance on the main characteristics of the atmospheric circulation under different forcing.

The paper needs for sure some editorial changes: the structure itself is not bad, but the story flow is very fragmented: the introduction seems a list of "this guy did this, this other guy did that" without really telling a story, being completely unbalanced and totally unfocused. The paper itself therefore resulted in a collection of results, but it is not really clear what authors are trying to prove.

Hence, I recommend a major revision.

The abstract is totally confusing. This sentence is completely out of context: "Under global warming the temperature gradients are expected to change: Enhanced warming is expected in the Arctic, largely near the surface, and at the equator at high altitudes, altering the meridional temperature gradient. Further, land-ocean contrasts will change due to enhanced land warming. Also there is a pronounced seasonality to these warming patterns."

The abstract needs substantial review, in order to reflect the text and highlight the main findings. Moreover, also from the abstract it is not really clear what authors want to prove.

Furthermore, authors insist to use inappropriate metrics for Hadley Circulation and jet-stream, making results very hard to compare with the wide literature about it. This is another weakness of the study.

Minor aesthetic adjustments are needed to some figures (see comments listed below).

Specific Comments:

Title: "...using a Statistical-Dynamical Atmosphere Model" Consider rephrasing as "...using a Statistical-Dynamical Atmospheric Model". Page 5 ln 20-21 "We change the temperature for each grid cell with respect to parameters for the three components in three steps. First, the parameter ðĬŚďTphi ...". I missed what is ðĬŚďTphi . I understood only it is a parameter. I guess you used a varying ðĬŚďTphi in order to have different sensitivities. Does ðĬŚďTphi vary in a range? Can you specify it here and not at page 6 ln 4,8? Page 6 ln 24: "...To obtain the strength of the jet stream for this analysis, we use seasonally (DJF)". It is not clear to me why you use specifically boreal winter season. Can you say something about it? Page 6 ln 25: "...for simplicity define the jet stream strength as the maximum of 〈ðĬŚć〉 between 10°N and 80°N at 9000 m height (corresponding to a pressure level of ca. 300 mbar). I like simplicity, but not oversimplifications. If you change the meridional temperature gradient, this affects 1) the position of the subtropical jet-stream (phi, plev), 2) on the strength of the subtropical jet-stream. Are you really sure that your metric effectively captures the maximum and the latitude of the jet-stream under different simulated "climates"? You should be able to say something about the choice of this specific metric and find a climate-invariant metric for the jet-stream. Page 7 ln 1-4: "metrics for HC". The way you describe the metrics is confusing. Are you computing the meridional mass stream-function? If yes, why don't you say it, rather then saying "by computing the integrated southward mass flux in the lower troposphere between 1000 mb and 500 mb from the zonal mean meridional wind velocity". As far as I understand this is equivalent to the classical stream-function by Oort and Yienger, 1996, Although you vertically integrated between 1000 hPa and 500 hPa. Be aware that 500 hPa is too shallow in order to get the full vertical extent of the HC, which actually it extends up to 200 hPa in deep tropics. I totally agree that there is confusion about metrics (I've read you reply to the reviewer from the previous submission) BUT, in order to get results comparable with literature and to avoid to introduce further confusion, you should take the vertical average of the stream-function between 400 hPa and 600 hPa, or 200 hPa and 700 hPa (usually it is a good practice to average first and stay above the boundary layer) and then calculate the width as the zero-crossing latitude. The integrated measure is not commonly used, therefore I do not recommend it. It can be potentially a wrong estimation of the HC strength. If you don't want to follow the literature, then you have to prove that YOUR metrics are consistent with the metrics used from the whole community. There is no easy way out about it. You might want to refer also to the first paper on metrics came out from a CLIVAR project on metrics for tropical width: https://www.geosci-model-dev-discuss.net/gmd-2018-124/ The TropD software package: Standardized methods for calculating Tropical Width Diagnostics By Ori Adam , Kevin M. Grise , Paul Staten , Isla R. Simpson , Sean M. Davis, Nicholas A. Davis , Darryn W. Waugh , and Thomas Birner. So, concluding, USE THE STANDARD METRICS FOR THE HADLEY CELL.

Page 7 ln 20: "In particular, the maximum strength, defined as the minimum between

the zero-crossings". This is really not clear to me. The strength is the max or the min value inside the NH or SH HC, respectively. Then the max strength is the max between NH and the SH poleward edges for the NH HC (if the NH HC is defined positive for clockwise overturning). This is also missing in the methods... Conventionally, the NH HC is positive, while the SH HC is negative. If you had written the equation for the stream-function this would have clarified explicitly. Please clarify.

Page 7 ln 20-22: "There exist bigger differences in the SH. This model bias might be related to the missing Antarctic ice sheet, upper-tropospheric ozone, the constant lapse rate assumption, or fundamental limitations of the equations." There is more than that. The cross-equatorial HC (e.g. the winter HC) is nearly inviscid limit. Therefore, its poleward extent and its strength are not dictated by eddy momentum flux (Schneider and Bordoni, 2008). At the same time, in the opposite hemisphere, the summer HC is dominated by eddy momentum flux divergence. Probably, the poor agreement in the SH is due to the statistical nature of the eddy representation in the SDAM. Therefore the use of the SDAM for HC analysis must consider only winter season. State it clearly. Page 11 ln24: "In this study, we observe a strengthening of storm track activity under increased global-mean temperature." The reference to the figure is missing. Provide it. Page 12 ln 28: "In our analysis the strengthening of the planetary waves depends on all temperature components. Larger meridional and zonal temperature asymmetries as well as global mean temperatures lead to stronger winds". The reference to the figure is missing. Provide it.

Figures:

Fig. 2 Caption: "Integrated northward mass flux in lower troposphere. ..." Please specify everywhere in the text that the winter you refer is the boreal winter. I have also some doubt about the magnitude. The conventional magnitude and unit for the atmospheric mass flux of the Northern Hemisphere psi_max_DJF is around 20 x 10^10 Kg/s or 200 Sverdrup. Why do you have here Kg/ s m2 and such weak values? In order to compare values with previous study it is warmly suggested to change the unit

according to the literature by performing the standard meridional mass stream-function (Oort and Yienger, 1996).

Fig. 5 Panel a) has not the same size of the others. Furthermore, x-labels of panels b,c,d are incorrect: it is supposed to be the latitude, right? Then, label correctly please. Please also write the unit of  (should be m/s, right?). Additionally, in panel a) you have pressure levels on the y-axis: why do you have altitude in meters for panels b,c,d? Please be consistent throughout panels. The same for Fig. 7.

Suggested literature:

Adam, O., Grise, K. M., Staten, P., Simpson, I. R., Davis, S. M., Davis, N. A., Waugh, D. W., and Birner, T.: The TropD software package: Standardized methods for calculating Tropical Width Diagnostics, Geosci. Model Dev. Discuss., https://doi.org/10.5194/gmd-2018-124, in review, 2018. Oort, A. H., & Yienger, J. J. (1996). Observed interannual variability in the Hadley circulation and its connection to ENSO.ÂăJournal of Climate,Âă9(11), 2751-2767. Schneider, T., & Bordoni, S. (2008). Eddy-mediated regime transitions in the seasonal cycle of a Hadley circulation and implications for monsoon dynamics.ÂăJournal of the Atmospheric Sciences,Âă65(3), 915-934.

---

## Referee Comment (RC2) · Anonymous Referee #2 · 31 Jul 2018

General Comments

In the present paper the authors use a statistical-dynamical model (Aeolus) to anal-yse the sensitivity of different components of the large scale atmospheric circulation (Hadley cell, jet stream, storm tracks, and planetary waves) to changes in surface temperature. They separate changes in the forcing temperature into global mean, meridional gradient, and zonal gradient. The results indicate a linear dependence of the strength of the Hadley cell, storm track activity and jets on global mean temperature and meridional gradient, with little sensitivity to zonal temperature asymmetries. Planetary waves appear to be sensitive to all three temperature components. The Hadley cell width shows a nonlinear dependence. The authors compare their findings with other studies.

In general, (i) intermediate complexity models, like the statistical-dynamical model used here, can be of great help investigating particular aspects of the climate system, (ii) a systematic analyses of the sensitivity of the global atmospheric circulation to changes in surface temperature can be an valuable contribution, and (iii) the components chosen by the authors are central to characterize the large scale circulation. Thus, in principle, overall concept and methodology of the study are sound. The paper is relatively well written and structured. However, unfortunately I do not feel that the work provides enough new and valuable information to warrant publication in the present form. So far, it is mostly an evaluation/validation of the Aeolus model illustrating that it shows similar sensitivities as more complex models (and observations). Thus, the study gives confidence to the model, but does not contribute much to the understanding of the climate system. The authors need to point out much clearer what is the particular aim (process, mechanism, etc.) they are focusing on (it seems like it is 'linearity' of response and/or sensitivity to individual forcing components), and, more important, what are new and significant findings which contribute to our understanding of the atmospheric circulation.

Specific Comments (random order)

1) Conclusions: So far, the central/only conclusion appears to be that the results serve as a validation of the model. This, as noted in General Comments, is insufficient to justify publication in my view. Instead, novel findings of the study need to be summarized, and their (potential) implications need to be discussed.

2) Eq.1: At P5L24/25 the authors state that using Eq.1 only the meridional temperature gradient is altered/updated in T1. Perhaps I got something wrong but as far as I understand Eq. 1 the non-zonal component is modified too. For example: for w_T_phi=0 all

temperatures (including, in particular, the zonal asymmetries) are the same as at the equator (=T_EQ(lambda)), and thus, in general, different from T_DJF(lambda). Please clarify.

3) Forcing: As far as I understand, and as it is stated in Sec. 3.2 and 7, the forcing of the simulations are surface temperatures for both land and ocean, but I'm still not sure: According to P4L23 the forcing appears to be sea level temperature (atmospheric temperatures extrapolated to sea level?), while in Sec. 3.1. L5/6 it is stated that the forcing is sea surface temperature only (and specific humidity at the surface). Finally, from the abstract one may infer that the forcing is the whole (3d) temperature field (P1L15-16). This may be homogenized/clarified to avoid confusions.

4) Stationary waves & topography: Since the authors exclude topographic influences (P4L20), I'm wondering if some modification of temperature is involved in regions with high topography (see also 3). In other words: would the model have stationary waves in a w_azonal=0 experiment?

5) Sensitivities: At various places the authors state that sensitivity to meridional gradient is larger than sensitivity to zonal asymmetries (e.g. P8L8/9). However, the authors apply relative change with respect to reference values (by changing the w's). I guess (though I'm not sure) the absolute values of the meridional gradient and of the zonal asymmetries differ, and I'm wondering whether this statement still holds if absolute changes are considered. In Sec. 4.2.3 (planetary waves) L11-15 it is not clear to me at all if relative of absolute changes are meant (i.e. w or the absolute values). Please clarify.

---

## Author Comment (AC1) · 3 Oct 2018

We thank the reviewer for the time she/he took and for the comments provided, which will help us to improve the manuscript. A pointwise reply to the reviewer's comment is given below.

**Major Comments:**

1.) *After the first rejection in ESDD, the new manuscript addressed pretty well the points made by reviewers in the previous submission. Authors now described very precisely the model and the model set-up used for the study, highlighting the strong points of the Statistical-dynamical Atmospheric Model SDAM-Aeolus 1.0 with respect to conventional CGMs used to investigate Hadley Cell (HC) dynamics and its changes. This represents the novelty of this study. However, it is very hard to find the main goal of the paper: in the beginning, disentangling the effect of changes in the meridional temperature gradient versus azonal temperature changes versus mean temperature changes on the boreal winter atmospheric circulation seems to be the main goal of the study. In the second part of the paper however, it seems that authors want to validate only the SDAM in order to assess its performance on the main characteristics of the atmospheric circulation under different forcing. The paper needs for sure some editorial changes: the structure itself is not bad, but the story flow is very fragmented: the introduction seems a list of "this guy did this, this other guy did that" without really telling a story, being completely unbalanced and totally unfocused. The paper itself therefore resulted in a collection of results, but it is not really clear what authors are trying to prove.*

    Thank you very much for the comment. As already written by the referee, the main goal of the paper is to investigate the effect of changes in the meridional temperature gradient versus azonal temperature changes versus mean temperature changes on the boreal winter atmospheric circulation. If possible, we compare the results with literature. Since most previous studies have analyzed only the combined effect of changes in several temperature components making a direct comparison difficult. We have rewritten the introduction and discussion to improve the readability.

2.) *The abstract is totally confusing. This sentence is completely out of context: "Under global warming the temperature gradients are expected to change: Enhanced warming is expected in the Arctic, largely near the surface, and at the equator at high altitudes, altering the meridional temperature gradient. Further, land-ocean contrasts will change due to enhanced land warming. Also there is a pronounced seasonality to these warming patterns." The abstract needs substantial review, in order to reflect the text and highlight the main findings. Moreover, also from the abstract it is not really clear what authors want to prove.*

    We agree with the reviewer and we have rewritten the abstract to:

Climate and weather conditions in the mid-latitudes are strongly driven by the large-scale atmosphere circulation. Observational data indicates that some components of the large-scale circulation have changed in recent decades, including the Hadley circulation, jet positions and storm tracks, but it remains unclear whether these changes are associated with greenhouse gas forcing or internal variability. Future climate simulations under high-emission scenarios show some robust changes in tropical and extra-tropical circulation but the uncertainties are large. Future simulations are characterized by enhanced warming in the Arctic at low levels and in the tropics at higher levels. In addition, land-ocean temperature contrasts are expected to change due to enhanced land-warming. The sensitivity of the large-scale circulation to these different changes in temperature gradients is not well quantified.

Here, we use a new statistical-dynamical atmosphere model (SDAM) to test the individual sensitivities of the large-scale atmospheric circulation to changes in the zonal temperature gradient, meridional temperature gradient and global-mean temperature. We analyse the Northern Hemisphere Hadley circulation, jet streams, storm tracks and planetary waves by systematically altering the zonal temperature asymmetry, the meridional temperature gradient, and the global mean temperature. Our results show that the strength of the Hadley cell, storm tracks and jet streams depends almost linearly on both the global mean temperature and the meridional temperature gradient whereas the zonal temperature asymmetry has little or no influence. The magnitude of planetary waves is affected by all three temperature components, as expected from theoretical dynamical considerations. The width of the Hadley cell behaves nonlinearly with respect to all three temperature components in the SDAM. Whether this is a model artefact or a true feature of tropical circulation should be assessed using high-resolution general circulation models.

3.) *Furthermore, authors insist to use inappropriate metrics for Hadley Circulation and jetstream, making results very hard to compare with the wide literature about it. This is another weakness of the study.*

We will use similar metrics as other literature used to compare the results: Integrated zonal wind velocity and maximum of the mean stream function from 3000m – 9000m.

**Specific Comments:**

1.) *Title: ": : :using a Statistical-Dynamical Atmosphere Model" Consider rephrasing as": : :using a Statistical-Dynamical Atmospheric Model".*

Thanks, we will change the title accordingly.

2.) *Page 5 ln 20-21 "We change the temperature for each grid cell with respect to parameters for the three components in three steps. First, the parameter dTphi : : :". I missed what is d'Tphi . I understood only it is a parameter. I guess you used a varying dTphi in order to have different sensitivities. dTphi vary in a range? Can you specify it here and not at page 6 ln 4,8?*

Thanks, the parameter $w_{T_\phi}$ is used to vary the meridional temperature gradient by cooling/warming the poles (p. 5, l. 21) and yes, we will write the analyzed range already in page 5.

3.) *Page 6 ln 24: ": : :To obtain the strength of the jet stream for this analysis, we use seasonally (DJF)". It is not clear to me why you use specifically boreal winter season. Can you say something about it?*

We meant that we use zonal mean zonal wind to analyse the strength of the jet stream. Since we analysed the sensitivity of the winter circulation, we used DJF averaged zonal mean zonal wind. We will remove this word, because it is already stated in the introduction.

4.) *Page 6 ln 25: ": : :for simplicity define the jet stream strength as the maximum of â ˇ Nl'ðˈI ´S c´âNˇ ł between 10 N and 80 N at 9000 m height (corresponding to a pressure level of ca. 300 mbar). I like plicity, but not oversimplifications. If you change the meridional temperature gradient, this affects 1) the position of the subtropical jet-stream (phi, plev), 2) on the strength of the subtropical jet-stream. Are you really sure that your metric effectively captures the maximum and the latitude of the jet-stream under different simulated "climates"? You should be able to say something about the choice of this specific metric and find a climate-invariant metric for the jet-stream.*

We agree with the referee and use instead the integrated wind velocity from 3000m – 9000m.

5.) *Page 7 ln 1-4: "metrics for HC". The way you describe the metrics is confusing. Are you computing the meridional mass streamfunction? If yes, why don't you say it, rather then saying "by computing the integrated southward mass flux in the lower troposphere between 1000 mb and 500 mb from the zonal mean meridional wind velocity". As far as I understand this is equivalent to the classical stream-function by Oort and Yienger, 1996, Although you vertically integrated between 1000 hPa and 500 hPa. Be aware that 500 hPa is too shallow in order to get the full vertical extent of the HC, which actually it*

*extends up to 200 hPa in deep tropics. I totally agree that there is confusion about metrics (I've read you reply to the reviewer from the previous submission) BUT, in order to get results comparable with literature and to avoid to introduce further confusion, you should take the vertical average of the stream-function between 400 hPa and 600 hPa, or 200 hPa and 700 hPa (usually it is a good practice to average first and stay above the boundary layer) and then calculate the width as the zero-crossing latitude. The integrated measure is not commonly used, therefore I do not recommend it. It can be potentially a wrong estimation of the HC strength. If you don't want to follow the literature, then you have to prove that YOUR metrics are consistent with the metrics used from the whole community. There is no easy way out about it. You might want to refer also to the first paper on metrics came out from a CLIVAR project on metrics for tropical width: https://www.geosci-model-devdiscuss.net/gmd-2018-124/ The TropD software package: Standardized methods for calculating Tropical Width Diagnostics By Ori Adam , Kevin M. Grise , Paul Staten , Isla R. Simpson , Sean M. Davis, Nicholas A. Davis , Darryn W. Waugh , and Thomas Birner. So, concluding, USE THE STANDARD METRICS FOR THE HADLEY CELL.*

Yes, it is the meridional mass stream function, we will rewrite the sentence. We will use the recommended metrics (mean of the 200hPa – 700hPa stream function).

6.) *Page 7 ln 20: "In particular, the maximum strength, defined as the minimum between Sthe zero-crossings". This is really not clear to me. The strength is the max or the min value inside the NH or SH HC, respectively. Then the max strength is the max between NH and the SH poleward edges for the NH HC (if the NH HC is defined positive for clockwise overturning). This is also missing in the methods: : : Conventionally, the NH HC is positive, while the SH HC is negative. If you had written the equation for the stream-function this would have clarified explicitly. Please clarify.*

*It is the maximal absolute value of the strength. In our case it is negative, since we consider the mass flux from 1000mb to 500mb and not 500mb to 1000mb.*

7.) *Page 7 ln 20-22: "There exist bigger differences in the SH. This model bias might be related to the missing Antarctic ice sheet, upper-tropospheric ozone, the constant lapse rate assumption, or fundamental limitations of the equations." There is more than that. The cross-equatorial HC (e.g. the winter HC) is nearly inviscid limit. Therefore, its poleward extent and its strength are not dictated by eddy momentum flux (Schneider and Bordoni, 2008). At the same time, in the opposite hemisphere, the summer HC is dominated by eddy momentum flux divergence. Probably, the poor agreement in the SH*

*is due to the statistical nature of the eddy representation in the SDAM. Therefore the use of the SDAM for HC analysis must consider only winter season. State it clearly.*

Thanks, we will include that as a possible explanation for the model bias.

8.) *Page 11 ln24: "In this study, we observe a strengthening of storm track activity under increased global-mean temperature." The reference to the figure is missing. Provide it.*

Yes, we will provide it (Fig. 8).

9.) *Page 12 ln 28: "In our analysis the strengthening of the planetary waves depends on all temperature components. Larger meridional and zonal temperature asymmetries as well as global mean temperatures lead to stronger winds". The reference to the figure is missing. Provide it.*

Thanks, we will provide it (Fig. 9).

**Figures:**

1.) *Fig. 2 Caption: "Integrated northward mass flux in lower troposphere. : : :" Please specify everywhere in the text that the winter you refer is the boreal winter. I have also some doubt about the magnitude. The conventional magnitude and unit for the atmospheric mass flux of the Northern Hemisphere psi_max_DJF is around 20 x 10ˆ10 Kg/s or 200 Sverdrup. Why do you have here Kg/ s m2 and such weak values? In order to compare values with previous study it is warmly suggested to change the unit. according to the literature by performing the standard meridional mass stream-function (Oort and Yienger, 1996).*

We will do as suggested.

[Figure]

**Figure 1 Integrated northward mass flux in lower troposphere. Black: Model output for w_(T_φ )=1 and w_azonal=1. Red: ERA Interim climatological boreal winter data.**

[Figure]

**Figure 2 Strength of the Hadley cell in dependence of w_(T_φ ) and w_azonal and ΔT_(G,PD). The arrow points in the direction of the strongest gradient in boreal winter.**

[Figure]

**Figure 3** Width of the Hadley cell in dependence of w_(T_φ ) and w_azonal and ΔT_(G,PD), whereby ΔT_(G,PD) is the difference between the present day temperature and the altered global mean temperature in boreal winter.

2.) *Fig. 5 Panel a) has not the same size of the others. Furthermore, x-labels of panels b,c,d are incorrect: it is supposed to be the latitude, right? Then, label correctly please. Please also write the unit of  (should be m/s, right?). Additionally, in panel a) you have pressure levels on the y-axis: why do you have altitude in meters for panels b,c,d? Please be consistent throughout panels. The same for Fig. 7.*

Thanks, we will correct it.

[Figure]

**Figure 4** Zonal mean zonal wind velocity $\overline{\langle u \rangle}$ representing the jet streams in boreal winter. The subpanel (a) shows ERA-Interim data, the others results from Aeolus with different $w_{T_\phi}$. In (b) $w_{T_\phi} = 0.75$, in (c) $w_{T_\phi} = 1.0$ and in (d) $w_{T_\phi} = 1.1$. All other parameters are set to standard values.

[Figure]

**Figure 5 Ed**dy kinetic energy representing storm track activity in boreal winter.. Panel (a) shows ERA Interim data, and (b)-(d) from Aeolus with different poleward temperature gradients $w_{T_\phi}$. In (b) $w_{T_\phi} = 0.75$, in (c) $w_{T_\phi} = 1.0$ and in (d) $w_{T_\phi} = 1.1$. All other parameters are set to climatology. With larger gradients the storm track acitivity gets stronger.

---

## Author Comment (AC2) · 3 Oct 2018

We thank the reviewer for the time she/he took and for the comments provided, which will help us to improve the manuscript. A pointwise reply to the reviewer's comment is given below.

**General Comments:**

1.) *In the present paper the authors use a statistical-dynamical model (Aeolus) to analyse the sensitivity of different components of the large scale atmospheric circulation (Hadley cell, jet stream, storm tracks, and planetary waves) to changes in surface temperature. They separate changes in the forcing temperature into global mean, meridional gradient, and zonal gradient. The results indicate a linear dependence of the strength of the Hadley cell, storm track activity and jets on global mean temperature and meridional gradient, with little sensitivity to zonal temperature asymmetries. Planetary waves appear to be sensitive to all three temperature components. The Hadley cell width shows a nonlinear dependence. The authors compare their findings with other studies. In general, (i) intermediate complexity models, like the statistical-dynamical model used here, can be of great help investigating particular aspects of the climate system, (ii) a systematic analyses of the sensitivity of the global atmospheric circulation to changes in surface temperature can be an valuable contribution, and (iii) the components chosen by the authors are central to characterize the large scale circulation. Thus, in principle, overall concept and methodology of the study are sound. The paper is relatively well written and structured. However, unfortunately I do not feel that the work provides enough new and valuable information to warrant publication in the present form. So far, it is mostly an evaluation/validation of the Aeolus model illustrating that it shows similar sensitivities as more complex models (and observations). Thus, the study gives confidence to the model, but does not contribute much to the understanding of the climate system. The authors need to point out much clearer what is the particular aim (process, mechanism, etc.) they are focusing on (it seems like it is 'linearity' of response and/or sensitivity to individual forcing components), and, more important, what are new and significant findings which contribute to our understanding of the atmospheric circulation.*

Thank you very much for this comment. We are happy that the reviewer agrees that this analysis is a sound approach. As written in response to referee 1, the main goal of the paper is to investigate the effect of changes in the meridional temperature gradient versus azonal temperature changes versus mean temperature changes on the boreal winter atmospheric circulation.

The novelty is the systematic approach. With this approach it is possible to scan the full temperature phase space. This way we can scan for 'non-linearities' in the system (i.e. the HCE might be very sensitive to dTdy only for a narrow range of dTdx values, and

outside of that range it is not sensitive). It is important to know such non-linearities as it could imply more abrupt changes under global warming.

In addition, we found:

- We find little of such non-linearities (most of the atmospheric circulation behaves in a linear fashion to thermal changes.
  - **Exception 1**: Planetary waves, which is well explained by theoretic dynamical consideration
  - **Exception 2**: Hadley cell edge, which could be a model artefact or a real feature – that should be tested with GCMs (as also written in the answer letter for referee 1)

**Specific Comments:**

1.) *Conclusions: So far, the central/only conclusion appears to be that the results serve as a validation of the model. This, as noted in General Comments, is insufficient to justify publication in my view. Instead, novel findings of the study need to be summarized, and their (potential) implications need to be discussed.*

We agree with the reviewer and will rewrite the conclusion to:

In this paper, we present a study on multiple fundamental components of the large-scale atmosphere dynamics to different surface temperature forcing with the statistical-dynamical Atmosphere model Aeolus 1.0. Due to the statistical-dynamical approach, Aeolus 1.0 is much faster than GCMs, which allows us to do 1000s of individual simulations and thus test the sensitivity of the dynamical fields to different surface temperature changes. This way one can disentangle and separately analyse the effect of global mean temperature, equator-to-pole temperature gradient and east-west temperature differences. Therefore, we are one of the first, who scan the full temperature phase space. This way we can scan for 'non-linearities' in the system (i.e. the Hadley cell edge might be very sensitive to meridional temperature gradients only for a narrow range of temperature gradient values, and outside of that range it is not sensitive). It is important to know such non-linearities as it could imply more abrupt changes under global warming. Exceptions are the planetary waves, which is well explained by theoretic dynamical consideration and the width of the Hadley, which could be a model artefact or a real feature. Latter should be tested with GCMs.

The model's climatology generally reproduces the dynamical fields of ERA-Interim, especially in the Northern Hemisphere, which is the focus of our analysis. If possible, we

compare our findings with results of the literature and conclude that most modelled changes are in line with theory and simulations.

These results also serve as an important validation of the dynamical core of the Aeolus. We could show that Aeolus is to our knowledge the first model that captures the dynamical interactions expected from dynamical principles between the large-scale circulation components of tropical circulation, jets, storm tracks and planetary waves. In future work we would like to use the gained knowledge to simulate only specific temperature component configurations to further explore the dependence of the different atmospheric large-scale circulations on near-surface temperature profiles.

2.) *Eq.1: At P5L24/25 the authors state that using Eq.1 only the meridional temperature gradient is altered/updated in T1. Perhaps I got something wrong but as far as I understand Eq. 1 the non-zonal component is modified too. For example: for w_T_phi=0 all paper temperatures (including, in particular, the zonal asymmetries) are the same as at the equator (=T_EQ(lambda)), and thus, in general, different from T_DJF(lambda). Please clarify.*

The parameter w_T_phi is for present day climatology values w_T_phi =1 and therefore T_DJF would be only occur for w_T_phi =1 (=100% present day climatology). Changing the parameter to w_T_phi=0 would mean that the gradient is 0 between equator and T_DJF and therefore T_DJF has to be T_EQ.

3.) *Forcing: As far as I understand, and as it is stated in Sec. 3.2 and 7, the forcing of the simulations are surface temperatures for both land and ocean, but I'm still not sure: According to P4L23 the forcing appears to be sea level temperature (atmospheric temperatures extrapolated to sea level?), while in Sec. 3.1. L5/6 it is stated that the forcing is sea surface temperature only (and specific humidity at the surface). Finally, from the abstract one may infer that the forcing is the whole (3d) temperature field (P1L15-16). This may be homogenized/clarified to avoid confusions.*

Thanks, it is atmospheric temperatures extrapolated to sea level. We will homogenize the other parts.

4.) *Stationary waves & topography: Since the authors exclude topographic influences (P4L20), I'm wondering if some modification of temperature is involved in regions with high topography (see also 3). In other words: would the model have stationary waves in a w_azonal=0 experiment?*

As written in comment 2, w_azonal=1 (=100% of present day climatology) is the present-day-climatology and therefore,  for w_azonal=0 there would be no stationary waves.

5.) *Sensitivities: At various places the authors state that sensitivity to meridional gradient is larger than sensitivity to zonal asymmetries (e.g. P8L8/9). However, the authors apply relative change with respect to reference values (by changing the w's).  I guess (though I'm not sure) the absolute values of the meridional gradient and of the zonal asymmetries differ,  and I'm wondering whether this statement still holds if absolute changes are considered.  In Sec.  4.2.3 (planetary waves) L11-15 it is not clear to me at all if relative of absolute changes are meant (i.e.  w or the absolute values). Please clarify.*

We mean relative changes and therefore you can be right. We will add that in the manuscript.

---

## Referee Report (RR1)

**Review of "Exploring the sensitivity of the large-scale atmosphere circulation to changes in surface temperature gradients using a Statistical-Dynamical Atmosphere Model "**

Sonja Totz, Stefan Petri , Jascha Lehmann, Erik Peukert, Dim Coumou
* * *
Reviewer #1

The paper aims to disentangle the role of global mean, meridional and azonal temperature changes on large scale atmospheric circulation (specifically focusing on jet stream, storm track, planetary wave and Hadley Cell) in the Northern Hemisphere. In order to separate the effect due to each other, the authors carried on simulations with a statistical-dynamical atmosphere model (SDAM) Aeolus 1.0. The authors found that the strength of the Hadley cell, storm tracks and jet streams depends almost linearly on both the global mean temperature and the meridional temperature gradient whereas the zonal temperature gradient has little or no influence. The width of the Hadley cell behaves nonlinearly with respect to all three temperature components.

After the first round of review authors improved substantially the manuscript. I found however some sources of ambiguities and some unclear statements.
I think that the paper can be accepted after a minor revision.
I have listed below my suggestions.

*Specific Comments:*

——————————————————————————————————————————-

Title: Consider re-title the paper as "Exploring the sensitivity of Northern Hemisphere atmospheric circulation to different surface temperature forcing using a Statistical Dynamical Atmospheric Model" as authors stated in conclusions (P.13 Ln 20-21 and P.14 Ln 1).

Abstract: Ln 14:…including the strength of the Hadley cell". Please include "…the strength and the width of the Hadley cell" since you are investigating both and both have changed in last decades.

Page 2, Ln 17-18: You can be more precise, and state why the HC is related to jet-streams, storm track and planetary wave activity. Please refer to the wide literature about it.

Page 3, Ln 3-4: "These waves strongly interact with storm track activity in the mid-latitudes".  Need reference here. Since you are investigating change in the atmospheric dynamics, you must further expand here the relationship between planetary waves and storm tracks. Few sentences would be enough.

Page 5, Ln 6: Large scale atmospheric circulation is surely related to large scale temperature difference but don't forget to mention that also angular momentum is a key ingredient. You can refer to this nice review:

https://agupubs.onlinelibrary.wiley.com/doi/10.1029/2006RG000213

Egger, J., Weickmann, K., & Hoinka, K. P. (2007). Angular momentum in the global atmospheric circulation. *Reviews of Geophysics*, *45*(4).

Page 4 Data and Methods: Maybe I missed it but I did not find any specification about the time-step of data you used to prescribe the SDAM. Are they monthly-seasonal mean or the seasonal mean are computed by daily means? I was just wondering if prescribing monthly mean data you can get a sufficient statistical representation of transient eddies and synoptic activity. I got from page 4 Ln 2 -3 that "Aeolus is based on time-averaged (over short time scales) equations… " but what are these short time scales? Furthermore, what is the temporal resolution of your model output? Please include these information.

Page 6, typo at ln 24 (after 700 mb).

Page 6, Ln 25 - 26: The strength and the width of the Hadley Cell are computed usually differently. The strength is the max or min inside each individual NH or SH cell described by the meridional mass streamfunction. The width is the zero-crossing latitude of the streamfunction averaged between 200 - 700 mb. Then there is an ambiguity in the definition of the strength and edges of the HC relative what you state in the response to reviewer. Please clarify.

Page 7, Ln 21 - 24. So: small difference of the meridional temperature gradient, narrower HC... Please refer to  Levine and Schneider, 2015 (page 2755 - Discussion).

Please refer in the discussion to this paper.

http://climate-dynamics.org/wp-content/uploads/2015/08/Levine-Schneider-2015.pdf

Levine, X. J., & Schneider, T. (2015). Baroclinic eddies and the extent of the Hadley circulation: An idealized GCM study. *Journal of the Atmospheric Sciences*, *72*(7), 2744-2761.

Page 7, Ln 26 - 27: there is a blank space or a discontinuity in the line. Remove.

Page 8, Ln 7: "… are visible" … rephrase as "… are detectable".

Page 8, Ln 14 - 16: You can expand here the description of the figures.

Page 9: Strength of the planetary waves. It is not really clear to me what you mean for "strength" here. Are you referring in change in the phase speed? Are you referring to meridional amplitude of Rossby waves? Can you be more precise please?

Page 9, Ln 25 - 26: I don't understand this statement. Why then in a warming climate the HC is robustly weakens while in the LGM it robustly strengthens?

Refer to Hill et al., 2015 also. Hill, S. A., Ming, Y., & Held, I. M. (2015). Mechanisms of forced tropical meridional energy flux change. *Journal of Climate*, *28*(5), 1725-1742.

Consider to rephrase.

Page 10, Ln 3 - 6: I don't understand this statement. Can you clarify?

Page 10, Ln 10: typo "present-day climate"

Page 12, "Strength of planetary waves" see comment above.

Page 13, Ln 26 - 27. Refer to Levine and Schneider paper.

Page 13, Lb 28 - 30: I don't understand this sentence. Please clarify.

————————————————————————————————————————-

Figures.

Fig. 5 and 7 : y-labels. It is pressure (mb), not altitude (km). Correct please.

---

## Author Response (AR2)

We thank the reviewer for the time she/he took and for the comments provided, which will help us to improve the manuscript. A pointwise reply to the reviewer's comment is given below.

1. *The paper aims to disentangle the role of global mean, meridional and azonal temperature changes on large scale atmospheric circulation (specifically focusing on jet stream, storm track, planetary wave and Hadley Cell) in the Northern Hemisphere. In order to separate the effect due to each other, the authors carried on simulations with a statistical-dynamical atmosphere model (SDAM) Aeolus 1.0. The authors found that the strength of the Hadley cell, storm tracks and jet streams depends almost linearly on both the global mean temperature and the meridional temperature gradient whereas the zonal temperature gradient has little or no influence. The width of the Hadley cell behaves nonlinearly with respect to all three temperature components. After the first round of review authors improved substantially the manuscript. I found however some sources of ambiguities and some unclear statements. I think that the paper can be accepted after a minor revision. I have listed below my suggestions.*

Thank you very much for this comment. We corrected all raising points as stated below.

*Specific Comments:*

1. *Title: Consider re-title the paper as "Exploring the sensitivity of Northern Hemisphere atmospheric circulation to different surface temperature forcing using a Statistical Dynamical Atmospheric Model" as authors stated in conclusions (P.13 Ln 20-21 and P.14 Ln 1).*

Thanks, we changed the title accordingly (p.1, l.1-3).

2. *Abstract: Ln 14:…including the strength of the Hadley cell". Please include "…the strength and the width of the Hadley cell" since you are investigating both and both have changed in last decades.*

We changed it (p. 1, l.14-15).

3. *Page 2, Ln 17-18: You can be more precise, and state why the HC is related to jetstreams, storm track and planetary wave activity. Please refer to the wide literature about it.*

We added the following paragraph (p. 2, l. 17-22):

Changes in the strength and the width of the Hadley cell circulation have strong implications for a variety of atmospheric phenomena such as jet streams, extratropical storms and planetary waves. The jet stream is caused by momentum conservation, when air of the Hadley cell moves poleward due to meridional contrasts in solar heating (Woollings, 2010). Therefore, changes of the Hadley cell directly influence the jet. Storm tracks are associated with fasttraveling Rossby waves that move from west to east, and there is a two-way interaction between jets and storm tracks. Storm tracks are both driven by the jet stream and can reinforce it (Totz et al., 2018).

4. *Page 3, Ln 3-4: "These waves strongly interact with storm track activity in the midlatitudes". Need reference here. Since you are investigating change in the atmospheric dynamics, you must further expand here the relationship between planetary waves and storm tracks. Few sentences would be enough.*

We added the following text (p. 3, l. 3 - 8):

Planetary waves are disturbances on longer time-scales (beyond 15 days) and are manifested by a displacement of the circumpolar flow out of zonal symmetry, as is apparent at higher latitudes. They lead to a gradual undulation of the jet stream about latitude circles (Salby, 2012). These planetary waves strongly interact with storm track activity in the mid-latitudes and impact the location and activity of the storm tracks (Branstator, 1994; Hoskins and Valdes, 1990). Especially the barotropic component of the planetary waves has a strong influence on the storm tracks and can change the distribution of storms by steering the disturbances (Branstator, 1994).

5. *Page 5, Ln 6: Large scale atmospheric circulation is surely related to large scale temperature difference but don't forget to mention that also angular momentum is a key ingredient. You can refer to this nice review:[…]*

Thanks, we included the paper reference (p. 3, l. 10).

6. *Page 4 Data and Methods: Maybe I missed it but I did not find any specification about the time-step of data you used to prescribe the SDAM. Are they monthly-seasonal mean or the seasonal mean are computed by daily means? I was just wondering if prescribing monthly mean data you can get a sufficient statistical representation of transient eddies and synoptic activity. I got from page 4 Ln 2 -3 that "Aeolus is based on time-averaged (over short time scales) equations… " but what are these short time scales? Furthermore, what is the temporal resolution of your model output? Please include these information.*

We calculated the seasonal mean data using monthly data as described in the section 3.1.

We added the following text (p. 5, l. 6 - 8):

In Aeolus stand-alone mode, the output is on the same time step as the input, i.e. seasonal means. However, internally, the time-stepping of the solver is one-hour resolution. In Aeolus, the synoptic wind fields u' and v' are parameterized in terms of the large-scale long-term fields (i.e. the seasonal mean fields).

Detailed information as comparison with reanalysis data is provided in Coumou et al. (2011).

7. Page 6, typo at ln 24 (after 700 mb).

Thanks, we corrected it (p. 7, l. 3).

8. *Page 6, Ln 25 - 26: The strength and the width of the Hadley Cell are computed usually differently. The strength is the max or min inside each individual NH or SH cell described by the meridional mass streamfunction. The width is the zero-crossing latitude of the streamfunction averaged between 200 - 700 mb. Then there is an ambiguity in the definition of the strength and edges of the HC relative what you state in the response to reviewer. Please clarify.*

We removed this sentence, because it belonged to the old HC width and magnitude calculation. Now we do it in the same way as suggested by the reviewer (and described in the first part).

9. *Page 7, Ln 21 - 24. So: small difference of the meridional temperature gradient, narrower HC... Please refer to Levine and Schneider, 2015 (page 2755 - Discussion).*

Thanks, we included it (p. 11, 16).

10. *Page 7, Ln 26 - 27: there is a blank space or a discontinuity in the line. Remove.*

We removed it.

11. *Page 8, Ln 7: "… are visible" … rephrase as "… are detectable".*

We rephrased as suggested (p. 8, l. 16).

12. *Page 8, Ln 14 - 16: You can expand here the description of the figures.*

We included the following text (p. 8, l. 22-26):
Fig. 5(b) and Fig. 5(d) show the impact of changes in the meridional temperature gradient $d\,T/d\phi$ on jet stream dynamics. With a higher meridional temperature gradient, the strength of the jet stream increases and with a lower temperature gradient the strength decreases indicated by the red color. For present day climatology values, the jet stream strength is about 25 m/s in the Northern Hemisphere. For a decrease of 10 % of the temperature gradient the velocity weakens to 20 m/s. With an increase of 10 % of the meridional temperature, the jet stream velocity is more than 30 m/s and with 10 per cent stronger in the Northern Hemisphere.

13. *Page 9: Strength of the planetary waves. It is not really clear to me what you mean for "strength" here. Are you referring in change in the phase speed? Are you referring to meridional amplitude of Rossby waves? Can you be more precise please?*

We were referring to the change of wind velocity. We rephrased it to wind velocity (p. 9, l. 9).

14. *Page 9, Ln 25 - 26: I don't understand this statement. Why then in a warming climate the HC is robustly weakens while in the LGM it robustly strengthens?*

We meant that the Hadley cell strength weakens by global warming due to enhanced latent heat release. However, the meridional temperature gradient has a stronger influence on the Hadley cell strength. We rewrote the sentence and included the reference (p. 10, l. 4 -7):

The strength of the Hadley cell depends strongly on the meridional temperature gradient with a stronger Hadley circulation for larger meridional gradient (Fig.4). Its strength is much less sensitive to global mean temperature. A rising global mean temperature leads to a weakening of the Hadley cell explainable by the enhanced latent heat release under warmer conditions. The specific humidity increases faster with temperature than precipitation and therefore the strength weakens (Hill et al., 2015).

15. *Page 10, Ln 3 - 6: I don't understand this statement. Can you clarify?*

Adam et al. analyzed the Hadley cell width to global mean temperature and global mean temperature. They got similar results for sensitivity of the Hadley cell width to the meridional temperature gradient but different results for the Hadley cell width to global mean temperature.

We rephrased the part to (p. 10, l.15-20):

Also Adam et al. (2014) examined the Hadley cell width using 6 reanalysis datasets, 22 Atmospheric Modeling Intercomparison Project (AMIP) simulations, and 11 historical Ocean-Atmosphere coupled simulations from phase 5 of the Climate Modeling Intercomparison Project (CMIP5). They observed the same behaviour for the meridional temperature gradient meaning that a weakening of the meridional temperature gradient leads to a widening of the Hadley cell width. The sensitivity of the Hadley cell width for global mean temperature shows in their analysis the opposite behaviour than in our analysis.

16. *Page 10, Ln 10: typo "present-day climate"*

We removed that sentence.

17. *Page 12, "Strength of planetary waves" see comment above.*

We rephrased it to wind velocity strength (p.13, l.4).

18. *Page 13, Ln 26 - 27. Refer to Levine and Schneider paper.*

We did as suggested (p. 14, l. 10).

19. *Page 13, Lb 28 - 30: I don't understand this sentence. Please clarify.*

This part referred to a later sentence and meant that we found some differences in our results compared to the literature :

If possible, we compare our findings with results of the literature and conclude that most modelled changes are in line with theory and simulations. Exceptions are the planetary waves, which are well explained by theoretic dynamical consideration and the width of the Hadley, which could be a model artefact or a real feature. Latter should be tested with GCMs.

*Specific Comments:*

1. *Fig. 5 and 7 : y-labels. It is pressure (mb), not altitude (km). Correct please.*

We corrected the labels.

[revised manuscript text omitted]